# Multi-objective Maximization of Monotone Submodular Functions with Cardinality Constraint

**Rajan Udwani**
Operations Research Center, M.I.T.
rudwani@alum.mit.edu

## Abstract

We consider the problem of multi-objective maximization of monotone submodular functions subject to cardinality constraint, often formulated as $\max_{|A|=k} \min_{i \in \{1,...,m\}} f_i(A)$. While it is widely known that greedy methods work well for a single objective, the problem becomes much harder with multiple objectives. In fact, Krause et al. (2008) showed that when the number of objectives $m$ grows as the cardinality $k$ i.e., $m = \Omega(k)$, the problem is inapproximable (unless $P = NP$). On the other hand, when $m$ is constant Chekuri et al. (2010) showed a randomized $(1 - 1/e) - \epsilon$ approximation with runtime (number of queries to function oracle) $n^{m/\epsilon^3}$.

We focus on finding a fast and practical algorithm that has (asymptotic) approximation guarantees even when $m$ is super constant. We first modify the algorithm of Chekuri et al. (2010) to achieve a $(1 - 1/e) - \epsilon$ approximation for $m = o(\frac{k}{\log^3 k})$, with $\epsilon \to 0$ as $k \to \infty$. This demonstrates a steep transition from constant factor approximability to inapproximability around $m = \Omega(k)$. Then using Multiplicative-Weight-Updates (MWU), we find a much faster $\tilde{O}(n/\delta^3)$ time asymptotic $(1 - 1/e)^2 - \delta$ approximation. While the above results are all randomized, we also give a simple deterministic $(1 - 1/e) - \epsilon$ approximation with runtime $kn^{m/\epsilon^4}$. Finally, we run synthetic experiments using Kronecker graphs and find that our MWU inspired heuristic outperforms existing heuristics.

## 1 Introduction

Several well known objectives in combinatorial optimization exhibit two common properties: the marginal value of any given element is non-negative and it decreases as more and more elements are selected. The notions of submodularity and monotonicity [1] nicely capture this property, resulting in the appearance of constrained monotone submodular maximization in a wide and diverse array of modern applications, including feature selection ([KG05, TCG+09]), network monitoring ([LKG+07]), news article recommendation ([EAVSG09]), sensor placement and information gathering ([OUS+08, GKS05, KGGK06, KLG+08]), viral marketing and influence maximization ([KKT03, HK16]), document summarization ([LB11]) and crowd teaching ([SB14]).

In this paper, we are interested in scenarios where multiple objectives, all monotone submodular, need to be simultaneously maximized subject to a cardinality constraint. This problem has an established line of work in both machine learning ([KMGG08]) and the theory community ([CVZ10]). Broadly speaking, there are two ways in which this paradigm has been applied:

**When there are several natural criteria that need to be simultaneously optimized:** such as in network monitoring, sensor placement and information gathering [OUS$^+$08, LKG$^+$07, KLG$^+$08, KMGG08]. For example in the problem of intrusion detection [OUS$^+$08], one usually wants to maximize the likelihood of detection while also minimizing the time until intrusion is detected, and the population affected by intrusion. The first objective is often monotone submodular and the latter objectives are monotonically decreasing supermodular functions [LKG$^+$07, KLG$^+$08]. Therefore, the problem is often formulated as an instance of cardinality constrained maximization with a small number of submodular objectives.

**When looking for solutions robust to the uncertainty in objective:** such as in feature selection [KMGG08, GR06], variable selection and experimental design [KMGG08], robust influence maximization [HK16]. In these cases, there is often inherently just a single submodular objective which is highly prone to uncertainty either due to dependence on a parameter that is estimated from data, or due to multiple possible scenarios that each give rise to a different objective. Therefore, one often seeks to optimize over the worst case realization of the uncertain objective, resulting in an instance of multi-objective submodular maximization. In some applications the number of objectives is given by the problem structure and can be larger even than the cardinality parameter. However, in applications such as robust influence maximization, variable selection and experimental design, the number of objectives is a design choice that trades off optimality with robustness.

## 1.1 Related Work

The problem of maximizing a monotone submodular function subject to a cardinality constraint,

$$P_0 := \max_{A \subseteq N, |A| \leq k} f(A),$$

goes back to the work of [NWF78, NW78], where they showed that the greedy algorithm gives a guarantee of $(1 - 1/e)$ and this is best possible in the value-oracle model. Later, [Fei98] showed that this is also the best possible approximation unless P=NP. While this settled the hardness and approximability of the problem, finding faster approximations remained an open line of inquiry. Notably, [BV14] found a faster algorithm for $P_0$ that improved the quadratic $O(nk)$ query complexity of the classical greedy algorithm to nearly linear complexity, by trading off on the approximation guarantee. This was later improved by [MBK$^+$15].

For the more general problem $\max_{A \in \mathcal{I}} f(A)$, where $\mathcal{I}$ is the collection of independent sets of a matroid; [CCPV11, Von08] in a breakthrough, achieved a $(1-1/e)$ approximation by (approximately) maximizing the *multilinear extension* of submodular functions, followed by suitable rounding. Based on this framework, tremendous progress was made over the last decade for a variety of different settings [CCPV11, Von08, FNS11, Von13, VCZ11, CVZ10, DV12].

In the multi-objective setting, [KMGG08] amalgamated various applications and formally introduced the following problem,

$$P_1 = \max_{A \subseteq N, |A| \leq k} \min_{i \in \{1,2,...,m\}} f_i(A),$$

where $f_i(.)$ is monotone submodular for every $i$. They call this the Robust Submodular Observation Selection (RSOS) problem and show that in general the problem is inapproximable unless $P = NP$. Consequently, they proceeded to give a bi-criterion approximation algorithm, called SATURATE, which achieves the optimal answer by violating the cardinality constraint. Note that their inapproximability result only holds when $m = \Omega(k)$. Another bi-criterion approximation was given more recently in [CLSS17].

On the other hand, [CVZ10] found a randomized $(1 - 1/e) - \epsilon$ approximation for constant $m$ in the more general case of matroid constraint, as an application of a new technique for rounding over a matroid polytope, called *swap rounding*. The runtime scales as $O(n^{m/\epsilon^3} + mn^8)$ [2]. Note, [CVZ10] consider a different but equivalent formulation of the problem that stems from the influential paper on multi-objective optimization [PY00]. The alternative formulation, which we introduce in Section 2, is the reason we call this a multi-objective maximization problem (same as [CVZ10]). For the special case of cardinality constraint (which will be our focus here), [OSU18] recently showed that the greedy algorithm can be generalized to achieve a *deterministic* $1 - 1/e - \epsilon$ approximation for the

special case of bi-objective maximization. Their runtime scales as $n^{1+1/\epsilon}$ and $\epsilon \leq 1/2$. To the best of our knowledge, when $m = o(k)$ no constant factor approximation algorithms or inapproximability results were known prior to this work.

## 1.2 Our Contributions

Our focus here is on the regime $m = o(k)$. This setting is essential to understanding the approximability of the problem for super-constant $m$ and includes several of the applications we referred to earlier. For instance, in network monitoring and sensor placement, the number of objectives is usually a small constant ([KMGG08, LKG$^+$07]). For robust influence maximization, the number of objectives depends on the underlying uncertainty but is often small ([HK16]). And in settings like variable selection and experimental design ([KMGG08]), where the number of objectives considered is a design choice. We show three algorithmic results with asymptotic approximation guarantees for $m = o(k)$.

**1. Asymptotically optimal approximation algorithm:** We give a $(1 - 1/e - \epsilon)(1 - \frac{m}{k\epsilon^3})$ approximation, which for $m = o\left(\frac{k}{\log^3 k}\right)$ and $\epsilon = \min\{\frac{1}{8 \ln m}, \sqrt[4]{\frac{m}{k}}\}$ tends to $1 - 1/e$ as $k \to \infty$. The algorithm is randomized and outputs such an approximation w.h.p. Observe that this implies a steep transition around $m$, due to the inapproximability result (to within any non-trivial factor) for $m = \Omega(k)$.

We obtain this via extending the algorithm of [CVZ10], which relies on the *continuous greedy* approach, resulting in a runtime of $\tilde{O}(mn^8)$. Note that there is no $\epsilon$ dependence in the runtime, unlike the result from [CVZ10]. The key idea behind the result is quite simple, and relies on exploiting the fact that we are dealing with a cardinality constraint, far more structured than matroids.

**2. Fast and practical approximation algorithm:** In practice, $n$ can range from tens of thousands to millions ([OUS$^+$08, LKG$^+$07]), which makes the above runtime intractable. To this end, we develop a fast $O(\frac{n}{\delta^3} \log m \log \frac{n}{\delta})$ time $(1 - 1/e)^2(1 - m/k\epsilon^3) - \epsilon - \delta$ approximation. Under the same asymptotic conditions as above, the guarantee simplifies to $(1 - 1/e)^2 - \delta$. We achieve this via the Multiplicative-Weight-Updates (MWU) framework, which replaces the bottleneck continuous greedy process. This costs us another factor of $(1 - 1/e)$ in the guarantee but allows us to leverage the runtime improvements for $P_0$ achieved in [BV14, MBK$^+$15].

MWU has proven to be a vital tool in the past few decades ([GK94, AK07, Bie06, Fle00, GK04, GK07, KY07, You95, You01, PST91, AHK12]). Linear functions and constraints have been the primary setting of interest in these works, but recent applications have shown its usefulness when considering non-linear and in particular submodular objectives ([AG12, CJV15]). Unlike these recent applications, we instead apply the MWU framework in vein of the Plotkin-Shmoys-Tardos scheme for linear programming ([PST91]), essentially showing that the non-linearity only costs us a another factor of $(1-1/e)$ in the guarantee and yields a nearly linear time algorithm. Independently, [CLSS17] applied the MWU framework in a similar manner and gave a new bi-criterion approximation. We further discuss how our result differs from theirs in Section 3.2.

**3. Finding a deterministic approximation for small $m$:** While the above results are all randomized, we also show a simple greedy based deterministic $1 - 1/e - \epsilon$ approximation with runtime $kn^{m/\epsilon^4}$. This follows by establishing an upper bound on the increase in optimal solution value as a function of cardinality $k$, which also resolves a weaker version of a conjecture posed in [OSU18].

**Outline:** We start with definitions and preliminaries in Section 2, where we also review relevant parts of the algorithm in [CVZ10] that are essential for understanding the results here. In Section 3, we state and prove the main results. Since the guarantees we present are asymptotic and technically converge to the constant factors indicated as $k$ becomes large, in Section 4 we test the performance of a heuristic closely inspired by our MWU based algorithm on Kronecker graphs [LCK$^+$10] of various sizes and find improved performance over previous heuristics even for small $k$ and large $m$.

## 2 Preliminaries

### 2.1 Definitions & review

We work with a ground set $N$ of $n$ elements and recall that we use $P_0$ to denote the single objective (classical) problem. [NWF78, NW78] showed that the natural *greedy algorithm* for $P_0$ summarized as, *starting with $\emptyset$, at each step add to the current set an element which adds the maximum marginal value until $k$ elements are chosen,* achieves a guarantee of $1 - 1/e$ for $P_0$ and that this is best possible. Formally, given set $A$ the marginal increase in value of function $f$ due to inclusion of set $X$ is given by, $f(X|A) = f(A \cup X) - f(A)$.

We use the notation $\mathbf{x}_S$ for the support vector of a set $S$ (1 along dimension $i$ if $i \in S$ and 0 otherwise). We also use the short hand $|\mathbf{x}|$ to denote the $\ell_1$ norm of $\mathbf{x}$. Given $f : 2^N \to \mathbb{R}$, recall that its *multilinear extension* over $\mathbf{x} = (x_1, \ldots, x_n) \in [0,1]^n$ is defined as, $F(\mathbf{x}) = \sum_{S \subseteq N} f(S) \prod_{i \in S} x_i \prod_{j \notin S} (1 - x_j)$. This function acts as a natural replacement for the original function $f$ in the *continuous greedy algorithm* ([CCPV11]). Like the greedy algorithm, the *continuous* version always moves in a feasible direction that best increases the value of function $F$. While evaluating the exact value of this function is naturally hard in general, for the purpose of using this function in optimization algorithms, approximations obtained using a sampling based oracle suffice ([BV14, CVZ10, CCPV11]). Given two vectors $\mathbf{x}, \mathbf{y} \in [0,1]^n$, let $\mathbf{x} \vee \mathbf{y}$ denote the component wise maximum. Then we define marginals for this function as, $F(\mathbf{x}|\mathbf{y}) = F(\mathbf{x} \vee \mathbf{y}) - F(\mathbf{y})$.

Now, we briefly discuss another formulation of the multi-objective maximization problem, call it $P_2$, introduced in [CVZ10]. In $P_2$ we are given a target value $V_i$ (positive real) with each function $f_i$ and the goal is to find a set $S^*$ of size at most $k$, such that $f_i(S^*) \geq V_i, \forall i \in \{1, \ldots, m\}$ or certify that no $S^*$ exists. More feasibly one aims to efficiently find a set $S \in \mathcal{I}$ such that $f_i(S) \geq \alpha V_i$ for all $i$ and some factor $\alpha$, or certify that there is no set $S^*$ such that $f_i(S^*) \geq V_i, \forall i$. Observe that w.l.o.g. we can assume $V_i = 1, \forall i$ (since we can consider functions $f_i(.)/V_i$ instead) and therefore $P_2$ is equivalent to the decision version of $P_1$: *Given $t > 0$, find a set $S^*$ of size at most $k$ such that $\min_i f_i(S^*) \geq t$, or give a certificate of infeasibility.*

When considering formulation $P_2$, *since we can always consider the modified submodular objectives* $\min\{f_i(.), V_i\}$, *we w.l.o.g. assume that $f_i(S) \leq V_i$ for every set $S$ and every function $f_i$*. Finally, for both $P_1, P_2$ we use $S_k$ to denote an optimal/feasible set (optimal for $P_1$, and feasible for $P_2$) to the problem and $OPT_k$ to denote the optimal solution value for formulation $P_1$. We now give an overview of the algorithm from [CVZ10] which is based on $P_2$. To simplify the description we focus on cardinality constraint, even though it is designed more generally for matroid constraint. We refer to it as **Algorithm 1** and it has three stages. Recall, the algorithm runs in time $O(n^{m/\epsilon^3} + mn^8)$.

**Stage 1:** Intuitively, this is a pre-processing stage with the purpose of picking a small initial set consisting of elements with 'large' marginal values, i.e. marginal value at least $\epsilon^3 V_i$ for some function $f_i$. This is necessary for technical reasons due to the rounding procedure in Stage 3.

Given a set $S$ of size $k$, fix a function $f_i$ and index elements in $S = \{s_1, \ldots, s_k\}$ in the order in which the greedy algorithm would pick them. There are at most $1/\epsilon^3$ elements such that $f_i(s_j|\{s_1, \ldots, s_{j-1}\}) \geq \epsilon^3 V_i$, since otherwise by monotonicity $f_i(S) > V_i$ (violating our w.l.o.g. assumption that $f_i(S) \leq V_i \forall i$). In fact, due to decreasing marginal values we have, $f_i(s_j|\{s_1, \ldots, s_{j-1}\}) < \epsilon^3 V_i$ for every $j > 1/\epsilon^3$. Therefore, we focus on sets of size $\leq m/\epsilon^3$ (at most $1/\epsilon^3$ elements for each function) to find an initial set such that the remaining elements have marginal value $\leq \epsilon^3 V_i$ for $f_i$, for every $i$. In particular, one can try all possible initial sets of this size (i.e. run subsequent stages with different starting sets), leading to the $n^{m/\epsilon^3}$ term in the runtime. Stages 2,3 have runtime polynomial in $m$ (in fact Stage 3 has runtime independent of $m$), hence Stage 1 is really the bottleneck. It is not obvious at all if one can do better than brute force enumeration over all possible starting sets and still retain the approximation guarantee, since the final solution must be an independent set of a matroid. However, as we show later, for cardinality constraints one can easily avoid enumeration.

**Stage 2:** Given a starting set $S$ from stage one, this stage works with the ground set $N - S$ and runs the continuous greedy algorithm. If a feasible set $S_k$ exists for the problem, then for the right starting set $S_1 \in S_k$, this stage outputs a fractional point $\mathbf{x}(k_1) \in [0,1]^n$ with $|\mathbf{x}(k_1)| = k_1 = k - |S|$ such that $F_i(\mathbf{x}(k_1)|\mathbf{x}_{N-S}) \geq (1 - 1/e - \epsilon)(V_i - f_i(S_1))$ for every $i$, where $\epsilon = 1/\Omega(k)$. The stage is

computationally expensive and takes time $\tilde{O}(mn^8)$. We refer the interested reader to [CVZ10] for further details (which will not be necessary for subsequent discussion).

**Stage 3:** For the right starting set $S_1$ (if one exists), Stage 2 successfully outputs a point $\mathbf{x}(k_1)$. Stage 3 now follows a random process that converts $\mathbf{x}(k_1)$ into a set $S_2$ of size $k_1$ such that, $S_2 \in N - S_1$ and $f_i(S_1 \cup S_2) \geq (1 - 1/e)(1 - \epsilon)V_i, \forall i$ as long as $\epsilon < 1/8 \ln m$. The rounding procedure is called *swap rounding* and we include a specialized version of the formal lemma below.

**Lemma 1.** *([CVZ10] Theorem 1.4, Theorem 7.2) Given $m$ monotone submodular functions $f_i(.)$ with the maximum value of singletons in $[0, \epsilon^3 V_i]$ for every $i$; a fractional point $\mathbf{x}$ and $\epsilon < \frac{1}{8\gamma \ln m}$. Swap Rounding yields a set $R$ with cardinality $|\mathbf{x}|$, such that,*

$$\sum_i \Pr[f_i(R) < (1 - \epsilon)F_i(\mathbf{x})] < me^{-1/8\epsilon} < 1/m^{\gamma-1}.$$

*Remark:* For any $\gamma > 1$, the above can be converted to a result w.h.p. by standard repetition. Also this is a simplified version of the matroid based result in [CVZ10].

## 3 Main Results

### 3.1 Asymptotic $(1 - 1/e)$ approximation for $m = o\left(\frac{k}{\log^3 k}\right)$

We replace the enumeration in Stage 1 with a single starting set, obtained by scanning once over the ground set. The main idea is simply that for the cardinality constraint case, any starting set that fulfills the Stage 3 requirement of small marginals will be acceptable (not true for general matroids).

**New Stage 1:** Start with $S_1 = \emptyset$ and pass over all elements once in arbitrary order. *For each element $e$, add it to $S_1$ if for some $i$, $f_i(e|S_1) \geq \epsilon^3 V_i$*. Note that we add at most $m/\epsilon^3$ elements (at most $1/\epsilon^3$ for each function). When the subroutine terminates, for every remaining element $e \in N \backslash S_1$, $f_i(e|S_1) < \epsilon^3 V_i, \forall i$ (as required by Lemma 1). Let $k_1 = k - |S_1|$ and note $k_1 \geq k - m/\epsilon^3$.

Stage 2 remains the same as Algorithm 1 and outputs a fractional point $\mathbf{x}(k_1)$ with $|\mathbf{x}(k_1)| = k_1$. Using basic properties of the multilinear extension and the continuous greedy framework we show for $\epsilon' = 1/\Omega(k)$,

$$F_i(\mathbf{x}(k_1)|\mathbf{x}_{S_1}) \geq \frac{k_1}{k}(1 - 1/e - \epsilon')(V_i - f_i(S_1)) \,\forall i. \tag{1}$$

The details are deferred to the supplementary material. Stage 3 rounds $\mathbf{x}(k_1)$ to $S_2$ of size $k_1$, and final output is $S_1 \cup S_2$.

**Theorem 2.** *For $\epsilon = \min\{\frac{1}{8 \ln m}, \sqrt[4]{\frac{m}{k}}\}$ we have, $f_i(S_1 \cup S_2) \geq (1 - \epsilon)(1 - 1/e)(1 - m/k\epsilon^3)V_i \,\forall i$ with constant probability. For $m = o\left(k/\log^3 k\right)$, the guarantee approaches $(1 - 1/e)$ asymptotically and the algorithm makes $\tilde{O}(mn^8)$ queries.*

*Proof.* From (1) and applying Lemma 1 we have, $f_i(S_2|S_1) \geq (1-\epsilon)(1-1/e-\epsilon')(1-m/k\epsilon^3)(V_i - f_i(S_1)), \forall i$. Therefore, $f_i(S_1 \cup S_2) \geq (1-\epsilon)(1-1/e-\epsilon')(1-m/k\epsilon^3)V_i, \forall i$. To refine the guarantee, we choose $\epsilon = \min\{\frac{1}{8 \ln m}, \sqrt[4]{\frac{m}{k}}\}$, where the $\frac{1}{8 \ln m}$ is due to Lemma 1 and the $\sqrt[4]{\frac{m}{k}}$ term is to balance $\epsilon$ and $m/k\epsilon^3$. Also $\epsilon' = 1/\Omega(k)$ therefore, the resulting guarantee becomes $(1 - 1/e)(1 - h(k))$, where the function $h(k) \to 0$ as $k \to \infty$, so long as $m = o\left(\frac{k}{\log^3 k}\right)$.

Note that the runtime is now independent of $\epsilon$. The first stage makes $O(mn)$ oracle queries, the second stage runs the continuous greedy algorithm on all functions simultaneously and makes $\tilde{O}(n^8)$ queries to each function oracle, contributing $O(mn^8)$ to the runtime. Stage 2 results in a fractional solution that can be written as a convex combination of $O(nk^2)$ sets of cardinality $k$ each (bases) (ref. Appendix A in [CCPV11]). For cardinality constraint, swap rounding can merge two bases in $O(k)$ time hence, the last stage takes time $O(nk^3)$. $\qquad\square$

### 3.2 Fast, asymptotic $(1 - 1/e)^2 - \delta$ approximation for $m = o\left(\frac{k}{\log^3 k}\right)$

While the previous algorithm achieves the best possible asymptotic guarantee, it is infeasible to use in practice. The main underlying issue was our usage of the continuous greedy algorithm in Stage 2

which has runtime $\tilde{O}(mn^8)$, but the flexibility offered by continuous greedy was key to maximizing the multilinear extensions of all functions at once. To improve the runtime we avoid continuous greedy and find an alternative in Multiplicative-Weight-Updates (MWU) instead. MWU allows us to combine multiple submodular objectives together into a single submodular objective and utilize fast algorithms for $P_0$ at every step.

The algorithm consists of 3 stages as before. Stage 1 remains the same as the New Stage 1 introduced in the previous section. Let $S_1$ be the output of this stage as before. Stage 2 is replaced with a fast MWU based subroutine that runs for $T = O(\frac{\ln m}{\delta^2})$ rounds and solves an instance of $SO$ during each round. Here $\delta$ is an artifact of MWU and manifests as a subtractive term in the approximation guarantee. The currently fastest algorithm for $SO$, in [MBK$^+$15], has runtime $O(n \log \frac{1}{\delta'})$ and an *expected* guarantee of $(1 - 1/e) - \delta'$. However, the slightly slower, but still nearly linear time $O(\frac{n}{\delta'} \log \frac{n}{\delta'})$ *thresholding* algorithm in [BV14], has (the usual) deterministic guarantee of $(1 - 1/e) - \delta'$. Both of these are known to perform well in practice and using either would lead to a runtime of $T \times \tilde{O}(n/\delta) = \tilde{O}(\frac{n}{\delta^3})$, which is a vast improvement over the previous algorithm.

Now, fix some algorithm $\mathcal{A}$ for $P_0$ with guarantee $\alpha$, and let $\mathcal{A}(f, k)$ denote the set it outputs given monotone submodular function $f$ and cardinality constraint $k$ as input. Note that $\alpha$ can be as large as $1 - 1/e$, and we have $k_1 = k - |S_1|$ as before. Then the new Stage 2 is,

---

**Algorithm 2** Stage 2: MWU

---

1: **Input:** $\delta, T = \frac{2 \ln m}{\delta^2}, \lambda_i^1 = 1/m, \tilde{f}_i(.) = \frac{f_i(.|S_1)}{V_i - f_i(S_1)}$
2: **while** $1 \le t \le T$ **do** $g^t(.) = \sum_{i=1}^{m} \lambda_i^t \tilde{f}_i(.)$
3: $X^t = \mathcal{A}(g^t, k_1)$
4: $m_i^t = \tilde{f}_i(X^t) - \alpha$
5: $\lambda_i^{t+1} = \lambda_i^t (1 - \delta m_i^t)$
6: $t = t + 1$
7: **Output:** $\mathbf{x}_2 = \frac{1}{T} \sum_{t=1}^{T} X^t$

---

*The point $\mathbf{x}_2$ obtained above is rounded to a set $S_2$ in Stage 3 (which remains unchanged). The final output is $S_1 \cup S_2$. Note that by abuse of notation we used the sets $X^t$ to also denote the respective support vectors. We continue to use $X^t$ and $\mathbf{x}_{X^t}$ interchangeably in the below.*

This application of MWU is unlike [AG12, CJV15], where broadly speaking the MWU framework is applied in a novel way to determine how an individual element is picked (or how a direction for movement is chosen in case of continuous greedy). In contrast, we use standard algorithms for $P_0$ and pick an entire set before changing weights. Also, [CJV15] uses MWU along with the continuous greedy framework to tackle harder settings, but for our setting using the continuous greedy framework eliminates the need for MWU altogether and in fact, we use MWU as a replacement for continuous greedy. Subsequent to our work we discovered a resembling application of MWU in [CLSS17]. Their application differs from Stage 2 above only in minor details, but unlike our result they give a bi-criterion approximation where the output is a set $S$ of cardinality up to $k \frac{\log m}{V \epsilon^2}$ such that $f_i(S) \ge (1 - 1/e - 2\epsilon)V$.

Now, consider the following intuitive schema. We would like to find a set $X$ of size $k$ such that $f_i(X) \ge \alpha V_i$ for every $i$. While this seems hard, consider the combination $\sum_i \lambda_i f_i(.)$, which is also monotone submodular for non-negative $\lambda_i$. We can easily find a set $X_\lambda$ such that $\sum_i \lambda_i f_i(X_\lambda) \ge \sum_i \lambda_i V_i$, since this is a single objective problem and we have fast approximations for $P_0$. However, for a fixed set of scalar weights $\lambda_i$, solving the $P_0$ problem instance need not give a set that has sufficient value for every individual function $f_i(.)$. This is where MWU comes into the picture. We start with uniform weights for functions, solve an instance of $P_0$ to get a set $X^1$. Then we change weights to undermine the functions for which $f_i(X^1)$ was closer to the target value and stress more on functions for which $f_i(X^1)$ was small, and repeat now with new weights. After running many rounds of this, we have a collection of sets $X^t$ for $t \in \{1, \dots, T\}$. Using tricks from standard MWU analysis ([AHK12]) along with submodularity and monotonicity, we show that $\sum_t \frac{f_i(X^t|S_1)}{T} \gtrsim (1 - 1/e)(V_i - f_i(S_1))$. Thus far, this resembles how MWU has been used in the literature for linear objectives, for instance the Plotkin-Shmoys-Tardos framework for solving LPs.

However, a new issue now arises due to the non-linearity of functions $f_i$. As an example, suppose that by some coincidence $\mathbf{x}_2 = \frac{1}{T}\sum_{t=1}^{T} X^t$ turns out to be a binary vector, so we easily obtain the set $S_2$ from $\mathbf{x}_2$. We want to lower bound $f_i(S_2|S_1)$, and while we have a good lower bound on $\sum_t \frac{f_i(X^t|S_1)}{T}$, it is unclear how the two quantities are related. More generally, we would like to show that $F_i(\mathbf{x}_2|\mathbf{x}_{S_1}) \geq \beta \sum_t \frac{f_i(X^t|S_1)}{T}$ and this would then give us a $\beta\alpha = \beta(1-1/e)$ approximation using Lemma 1. Indeed, we show that $\beta \geq (1-1/e)$, resulting in a $(1-1/e)^2$ approximation. In the lemmas that follow, we state this more concretely (proofs deferred to supplementary material).

**Lemma 3.** $g^t(X^t) \geq \frac{k_1}{k}\alpha\sum_i \lambda_i^t, \forall t.$

**Lemma 4.**
$$\frac{\sum_t \tilde{f}_i(X^t)}{T} \geq \frac{k_1}{k}(1-1/e) - \delta \,, \forall i.$$

**Lemma 5.** *Given monotone submodular function $f$, its multilinear extension $F$, sets $X^t$ for $t \in \{1, \ldots, T\}$, and a point $\mathbf{x} = \sum_t X^t/T$, we have,*
$$F(\mathbf{x}) \geq (1-1/e)\frac{1}{T}\sum_{t=1}^{T} f(X^t).$$

**Theorem 6.** *For $\epsilon = \min\{\frac{1}{8\ln m}, \sqrt[4]{\frac{m}{k}}\}$, the algorithm makes $O(\frac{n}{\delta^3}\log m \log \frac{n}{\delta})$ queries, and with constant probability outputs a feasible $(1-\epsilon)(1-1/e)^2(1-\frac{m}{k\epsilon^3})-\delta$ approximate set. Asymptotically, $(1-1/e)^2 - \delta$ approximate for $m = o(k/\log^3 k)$.*

*Proof.* Combining Lemmas 4 & 5 we have, $\tilde{F}_i(\mathbf{x}_2) \geq (1-1/e)\frac{\sum_t \tilde{f}_i(X^t)}{T} \geq \frac{k_1}{k}(1-1/e)^2 - \delta \,, \forall i.$ The asymptotic result follows just as in Theorem 2. For runtime, note that Stage 1 takes time $O(n)$. Stage 2 runs an instance of $\mathcal{A}(.)$, $T$ times, leading to an upper bound of $O((\frac{n}{\delta}\log\frac{n}{\delta}) \times \frac{\log m}{\delta^2}) = O(\frac{n}{\delta^3}\log m \log \frac{n}{\delta})$, if we use the thresholding algorithm in [BV14] (at the cost of a multiplicative factor of $(1-\delta)$ in the approximation guarantee). Finally, swap rounding proceeds in $T$ rounds and each round takes $O(k)$ time, leading to total runtime $O(\frac{k}{\delta^2}\log m)$ for Stage 3. Combining all three we get a runtime of $O(\frac{n}{\delta^3}\log m \log \frac{n}{\delta})$. □

### 3.3 Variation in optimal solution value and derandomization

Consider the problem $P_0$ with cardinality constraint $k$. Given an optimal solution $S_k$ with value $OPT_k$ for the problem, it is not difficult to see that for arbitrary $k' \leq k$, there is a subset $S_{k'} \subseteq S_k$ of size $k'$, such that $f(S_{k'}) \geq \frac{k'}{k}OPT_k$. For instance, indexing the elements in $S_k$ using the greedy algorithm, and choosing the set given by the first $k'$ elements gives such a set. This implies $OPT_{k'} \geq \frac{k'}{k}OPT_k$, and the bound is easily seen to be tight.

This raises a natural question: Can we generalize this bound on variation of optimal solution value with varying $k$, for multi-objective maximization? A priori, this isn't obvious even for modular functions. In particular, note that indexing elements in order they are picked by the greedy algorithm doesn't suffice since there are many functions and we need to balance values amongst all. We show that one can indeed derive such a bound (proof in supplementary material).

**Lemma 7.** *Given that there exists a set $S_k$ such that $f_i(S_k) \geq V_i, \forall i$ and $\epsilon < \frac{1}{8\ln m}$. For every $k' \in [m/\epsilon^3, k]$, there exists $S_{k'} \subseteq S_k$ of size $k'$, such that,*
$$f_i(S_{k'}) \geq (1-\epsilon)\Big(\frac{k'-m/\epsilon^3}{k-m/\epsilon^3}\Big)V_i, \forall i.$$

**Conjecture in [OSU18]:** Note that this resolves a slightly weaker version of the conjecture in [OSU18] for constant $m$. The original conjecture states that for constant $m$ and every $k' \geq m$, there exists a set $S$ of size $k'$, such that $f_i(S) \geq \frac{k'-\Theta(1)}{k}V_i, \forall i$. Asymptotically, both $\frac{k'-m/\epsilon^3}{k-m/\epsilon^3}$ and $\frac{k'-\Theta(1)}{k}$ tend to $\frac{k'}{k}$. This implies that for large enough $k'$, we can choose sets of size $k'$ ($k'$-tuples) at each step to get a deterministic (asymptotically) $(1-1/e) - \epsilon$ approximation with runtime $O(kn^{m/\epsilon^4})$ for the multi-objective maximization problem, when $m$ is constant (all previously known approximation algorithms, as well as the ones presented earlier, are randomized). We defer the proof to supplementary material.

**Theorem 8.** *For $k' = \frac{m}{\epsilon^4}$, choosing $k'$-tuples greedily w.r.t. $h(.) = \min_i f_i(.)$ is asymptotically $(1 - 1/e)(1 - 2\epsilon)$ approximate, while making $kn^{m/\epsilon^4}$ queries.*

## 4 Experiments on Kronecker Graphs

We choose synthetic experiments where we can control the parameters to see how the algorithm performs in various scenarios, esp. since we would like to test how the MWU algorithm performs for small values of $k$ and $m = \Omega(k)$. We work with formulation $P_1$ of the problem and consider a multi-objective version of the *max-k-cover* problem on graphs. Random graphs for our experiments were generated using the Kronecker graph framework introduced in [LCK+10]. These graphs exhibit several natural properties and are considered a good approximation for real networks (esp. social networks [HK16]).

We compare three algorithms: (i) A baseline greedy heuristic, labeled GREEDY, which focuses on one objective at a time and successively picks $k/m$ elements greedily w.r.t. each function (formally stated in supplementary material). (ii) A bi-criterion approximation called SATURATE from [KMGG08], to the best of our knowledge this is considered state-of-the-art for the problem. (iii) We compare these algorithms to a heuristic inspired by our MWU algorithm. This heuristic differs from the algorithm discussed earlier in two ways. Firstly, we eliminate Stage 1 which was key for technical analysis but in practice makes the algorithm perform similar to GREEDY. Second, instead of simply using the the swap rounded set $S_2$, we output the best set out of $\{X^1, \ldots, X^T\}$ and $S_2$. Also, for both SATURATE and MWU we estimate target value $t$ using binary search and consider capped functions $\min\{f_i(.), t\}$. Also, for the MWU stage, we tested $\delta = 0.5$ or $0.2$.

---

**Algorithm 3** GREEDY
1: **Input:** $k, m, f_i(.)$ for $i \in [m]$
2: $S = \emptyset, i = 1$
3: **while** $|S| \leq k - 1$ **do**
4: $S = S + \arg\max_{x \in N-S} f_i(x|S)$
5: $i = i + 1 \mod m + 1$
6: **Output:** $S$

---

---

**Algorithm 4** SATURATE
1: Input: $k, t, f_1, \ldots, f_m$ and set $A = \emptyset$
2: $g(.) = \sum_i \min\{f_i(.), t\}$
3: **while** $|A| < k$ **do** $A = A + \arg\max_{x \in N-A} g(x|A)$
4: Output: $A$

---

We pick Kronecker graphs of sizes $n \in \{64, 512, 1024\}$ with random initiator matrix [3] and for each $n$, we test for $m \in \{10, 50, 100\}$. Note that each graph here represents an objective, so for a fixed $n$, we generate $m$ Kronecker graphs to get $m$ *max-cover* objectives. For each setting of $n, m$ we evaluate the solution value for the heuristics as $k$ increases and show the average performance over 30 trials for each setting. All experiments were done using MATLAB.

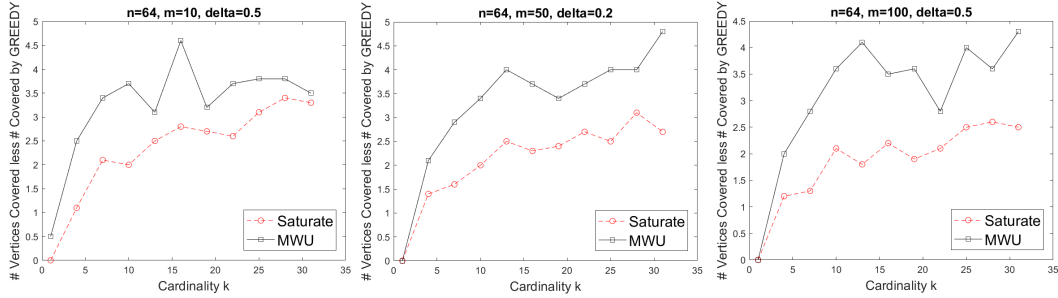

Figure 1: Plots for graphs of size 64. Number of objectives increases from left to right. The X axis is the cardinality parameter $k$ and Y axis is difference between # vertices covered by MWU and SATURATE minus the # vertices covered by GREEDY for the same $k$. MWU outperforms the other algorithms in all cases, with a max. gain (on SATURATE) of 9.80% for $m = 10$, 12.14% for $m = 50$ and 16.12% for $m = 100$.

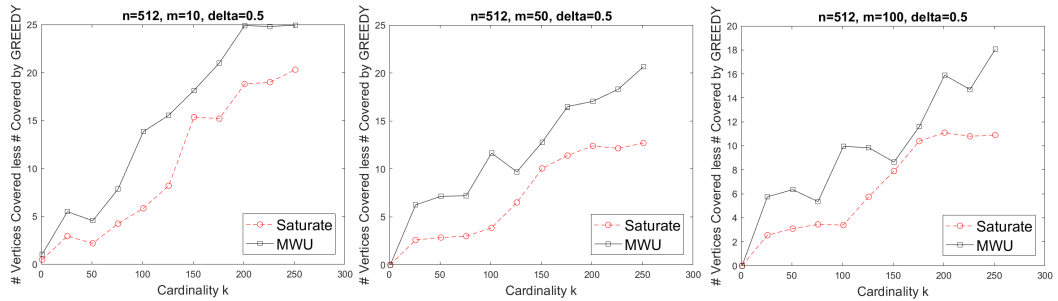

Figure 2: Plots for graphs of size 512. MWU outperforms SATURATE in all cases with a max. gain (on SATURATE) of 7.95% for $m = 10$, 10.08% for $m = 50$ and 10.01% for $m = 100$.

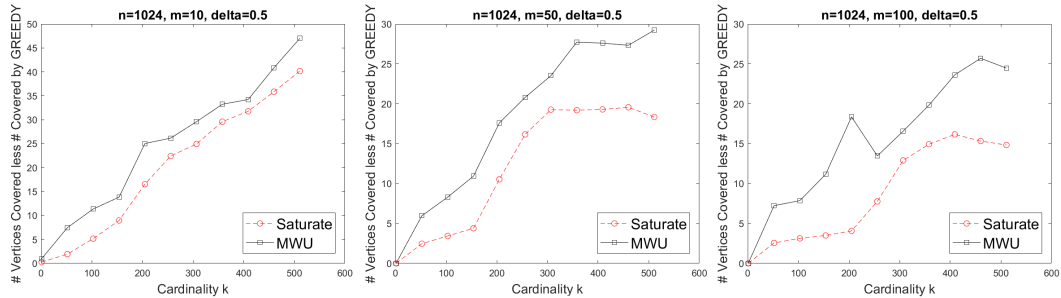

Figure 3: Plots for graphs of size 1024. MWU outperforms SATURATE in all cases, with max. gain (on SATURATE) of 6.89% for $m = 10$, 5.02% for $m = 50$ and 7.4% for $m = 100$.

## 5    Open Problems

A natural open question here is whether one can achieve similar approximations for a general matroid constraint. Additionally, it also of interest to ask if there are fast algorithms with guarantee closer to $1 - 1/e$, in contrast to the guarantee of $(1 - 1/e)^2$ shown here. Further, it is unclear if one can extend the results right up to $m = o(k)$.

**Acknowledgments**

The author gratefully acknowledges partial support from ONR Grant N00014-17-1-2194. The author would also like to thank James B. Orlin and anonymous referees for their insightful comments and feedback on early drafts of this work.

## Footnotes

[1] A set function $f : 2^N \to \mathbb{R}$ on the ground set $N$ is called submodular when $f(A + a) - f(A) \le f(B + a) - f(B)$ for all $B \subseteq A \subseteq N$ and $a \in N \setminus A$.. The function is monotone if $f(B) \le f(A)$ for all $B \subseteq A$. We assume $f(\emptyset) = 0$, then due to monotonicity we have that $f$ is non-negative.

[2]The $n^8$ term could potentially be improved to $n^5$ by leveraging subsequent work [BV14, FW14].

[3]To generate a Kronecker graph one needs a small initiator matrix. Using [LCK+10] as a guideline we use random matrices of size $2 \times 2$, each entry chosen uniformly randomly (and independently) from $[0, 1]$. Matrices with sum of entries smaller than 1 are discarded to avoid highly disconnected graphs.

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
