[Supplementary Material]

# Supplementary Material: Multi-objective Maximization of Monotone Submodular Functions with Cardinality Constraint

## 1 Some More Notation and Preliminaries

Let $\beta(\eta) = 1 - \frac{1}{e^\eta} \in [0, 1 - 1/e]$ for $\eta \in [0, 1]$. Note that $\beta(1) = (1 - 1/e)$. Further, for $k' \le k$,

$$\beta(k'/k) = (1 - e^{1-k'/k}/e) \ge (1 - 1/e)k'/k. \tag{1}$$

This function appears naturally in our analysis and will be useful for expressing approximation guarantees. Next, the lemma below formalizes Stage 2 of the algorithm in [CVZ10].

**Lemma 8.** *([CVZ10] Lemma 7.3) Given submodular functions $f_i$ and values $V_i$, cardinality $k$, the continuous greedy algorithm finds a point $\mathbf{x} \in [0, 1]^n$ such that $F_i(\mathbf{x}(k)) \ge (1 - 1/e - \epsilon')V_i \, \forall i$ with $\epsilon' = 1/\Omega(k)$, or outputs a certificate of infeasibility.*

## 2 Missing Proofs from Section 3.1

**Corollary 9.** *Given a point $\mathbf{x} \in [0, 1]^n$ with $|\mathbf{x}| = k$ and a multilinear extension $F$ of a monotone submodular function, for every $k_1 \le k$,*

$$F\left(\frac{k_1}{k}\mathbf{x}\right) \ge \frac{k_1}{k}F(\mathbf{x}).$$

*Proof.* Note that the statement is true for concave $F$. The proof now follows directly from the concavity of multilinear extensions in positive directions (Section 2.1 of [CCPV11]). □

**Lemma 10.** $F_i(\mathbf{x}(k_1)|\mathbf{x}_{S_1}) \ge (\beta(1) - \epsilon')\frac{k_1}{k}(V_i - f_i(S_1))$ *for every $i$.*

*Proof.* Recall that $S_k$ denotes a feasible solution with cardinality $k$, and let $\mathbf{x}_{S_k}$ denote its characteristic vector. Clearly, $|\mathbf{x}_{S_k \setminus S_1}| \le k$ and $F_i(\mathbf{x}_{S_k \setminus S_1}|\mathbf{x}_{S_1}) = f_i(S_k|S_1) \ge (V_i - f_i(S_1))$ for very $i$. And now from Corollary 9, we have that there exists a point $\mathbf{x}'$ with $|\mathbf{x}'| = k_1$ such that $F_i(\mathbf{x}'|\mathbf{x}_{S_1}) \ge \frac{k_1}{k}F_i(\mathbf{x}_{S_k \setminus S_1}|\mathbf{x}_{S_1})$ for every $i$. Finally, using Lemma 8 we have $F_i(\mathbf{x}(k_1)|\mathbf{x}_{S_1}) \ge (\beta(1) - \epsilon')F_i(\mathbf{x}'|S_1)$, which gives the desired bound. □

## 3 Missing Proofs from Section 3.2

**Lemma 3.** $g^t(X^t) \ge \frac{k_1}{k}\alpha \sum_i \lambda_i^t, \forall t.$

*Proof.* Consider the optimal set $S_k$ and note that $\sum_i \lambda_i^t \tilde{f}_i(S_k) \ge \sum_i \lambda_i^t, \forall t$. Now the function $g^t(.) = \sum_i \lambda_i^t \tilde{f}_i(.)$, being a convex combination of monotone submodular functions, is also monotone submodular. We would like to show that there exists a set $S'$ of size $k_1$ such that $g^t(S') \ge \frac{k_1}{k}\sum_i \lambda_i^t$. Then the claim follows from the fact that $\mathcal{A}$ is an $\alpha$ approximation for monotone submodular maximization with cardinality constraint.

To see the existence of such a set $S'$, greedily index the elements of $S_k$ using $g^t(.)$. Suppose that the resulting order is $\{s_1, \ldots, s_k\}$, where $s_i$ is such that $g^t(s_i|\{s_1, \ldots, s_{i-1}\}) \ge g^t(s_j|\{s_1, \ldots, s_{i-1}\})$ for every $j > i$. Then the truncated set $\{s_1, \ldots, s_{k-|S_1|}\}$ has the desired property, and we are done. □

**Lemma 4.**
$$\frac{\sum_t \tilde{f}_i(X^t)}{T} \ge \frac{k_1}{k}(1 - 1/e) - \delta \, , \forall i.$$

*Proof.* Suppose we have,

$$\frac{\sum_t \tilde{f}_i(X^t) - \alpha}{T} + \delta \ge \frac{1}{T}\sum_t \sum_i \frac{\lambda_i^t}{\sum_i \lambda_i^t}(\tilde{f}_i(X^t) - \alpha), \forall i. \tag{2}$$

Then assuming $\alpha = (1 - 1/e)$, the RHS above simplifies to,

$$\frac{1}{T}\sum_t \frac{g(X^t)}{\sum_i \lambda_i^t} - (1 - 1/e) \quad\geq (1 - 1/e)(\frac{k_1}{k} - 1) \quad \text{(using Lemma 3)}$$

And we have for every $i$,

$$\frac{\sum_t \tilde{f}_i(X^t) - (1 - 1/e)}{T} + \delta \quad\geq (1 - 1/e)\frac{k_1}{k} - 1)$$

$$\frac{\sum_t \tilde{f}_i(X^t)}{T} \quad\geq \frac{k_1}{k}(1 - 1/e) - \delta.$$

Now, the proof for (2) closely resembles the analysis in Theorem 3.3 and 2.1 in (**author?**) AHK12. We will use the potential function $\Phi^t = \sum_i \lambda_i^t$. Let $p_i^t = \lambda_i^t/\Phi^t$ and $M^t = \sum_i p_i^t m_i^t$. Then we have,

$$\Phi^{t+1} \quad= \sum_i \lambda_i^t(1 - \delta m_i^t)$$

$$= \Phi^t - \delta\Phi^t \sum_i p_i^t m_i^t$$

$$= \Phi^t(1 - \delta M^t) \leq \Phi^t e^{-\delta M^t}$$

After $T$ rounds, $\Phi^T \leq \Phi^1 e^{-\delta\sum_t M^t}$. Further, for every $i$,

$$\Phi^T \geq w_i^T = \frac{1}{m}\prod_t(1 - \delta m_i^t)$$

$$\ln(\Phi^1 e^{-\delta\sum_t M^t}) \geq \sum_t \ln(1 - \delta m_i^t) - \ln m$$

$$\delta\sum_t M^t \leq \ln m + \sum_t \ln(1 - \delta m_i^t)$$

Using $\ln(\frac{1}{1-\epsilon}) \leq \epsilon + \epsilon^2$ and $\ln(1 + \epsilon) \geq \epsilon - \epsilon^2$ for $\epsilon \leq 0.5$, and with $T = \frac{2\ln m}{\delta^2}$ and $\delta < (1 - 1/e)$ (for a positive approximation guarantee), we have,

$$\frac{\sum_t M^t}{T} \leq \delta + \frac{\sum_t m_i^t}{T}, \forall i.$$

$\square$

**Lemma 5.** *Given monotone submodular function $f$, its multilinear extension $F$, sets $X^t$ for $t \in \{1, \ldots, T\}$, and a point $\mathbf{x} = \sum_t X^t/T$, we have,*

$$F(\mathbf{x}) \geq (1 - 1/e)\frac{1}{T}\sum_{t=1}^T f(X^t).$$

*Proof.* Consider the concave closure of a submodular function $f$,

$$f^+(\mathbf{x}) = \max_\alpha\{\sum_X \alpha_X f(X)| \sum_X \alpha_X X = \mathbf{x}, \sum_X \alpha_X \leq 1, \alpha_X \geq 0\,\forall X \subseteq N\}.$$

Clearly, $f_i^+(\mathbf{x}) \geq \frac{\sum_t f_i(X^t)}{T}$. So it suffices to show $F_i(\mathbf{x}) \geq (1 - 1/e)f_i^+(\mathbf{x})$, which in fact, follows from Lemmas 4 and 5 in [CCPV07].

Alternatively, we now give a novel and direct proof for the statement. We abuse notation and use $\mathbf{x}_{X^t}$ and $X^t$ interchangeably. Let $\mathbf{x} = \sum_{t=1}^T X^t/T$ and w.l.o.g., assume that sets $X^t$ are indexed such that $f(X^j) \geq f(X^{j+1})$ for every $j \geq 1$. Further, let $f(X^t)/T = a^t$ and $\sum_t a^t = A$.

Recall that $F(\mathbf{x})$ can be viewed as the expected function value of the set obtained by independently sampling element $j$ with probability $x_j$. Instead, consider the alternative random process where starting with $t = 1$, one samples each element in set $X^t$ independently with probability $1/T$. The random process runs in $T$ steps and the probability of an element $j$ being chosen at the end of the process is exactly $p_j = 1 - (1 - 1/T)^{Tx_j}$, independent of all other elements. Let $\mathbf{p} = (p_1, \ldots, p_n)$, it follows that the expected value of the set sampled using this process is given by $F(\mathbf{p})$. Observe that for every $j$, $p_j \leq x_j$ and therefore, $F(\mathbf{p}) \leq F(\mathbf{x})$. Now in step $t$, suppose the newly sampled

subset of $X^t$ adds marginal value $\Delta^t$. From submodularity we have, $\mathbb{E}[\Delta^1] \geq \frac{f(X^1)}{T} = a^1$ and in general, $\mathbb{E}[\Delta^t] \geq \frac{f(X^t) - \mathbb{E}[\sum_{j=1}^{t-1} \Delta_j]}{T} \geq a^t - \frac{1}{T}\sum_{j=1}^{t-1} \mathbb{E}[\Delta^j]$.

To see that $\sum_t \mathbb{E}[\Delta^t] \geq (1 - 1/e)A$, consider a LP where the objective is to minimize $\sum_t \gamma^t$ subject to $b^1 \geq b^2 \cdots \geq b^T \geq 0$; $\sum b^t = A$ and $\gamma^t \geq b^t - \frac{1}{T}\sum_{j=1}^{t-1} \gamma^j$ with $\gamma^0 = 0$. Here $A$ is a parameter and everything else is a variable. Observe that the extreme points are characterized by $j$ such that, $\sum b^t = A$ and $b^t = b^1$ for all $t \leq j$ and $b^{j+1} = 0$. For all such points, it is not difficult to see that the objective is at least $(1 - 1/e)A$. Therefore, we have $F(\mathbf{p}) \geq (1 - 1/e)A = (1 - 1/e)\sum_t f(X^t)/T$, as desired.

$\square$

## 4 Missing Proofs from Section 3.3

**Lemma 7.** *Given that there exists a set $S_k$ such that $f_i(S_k) \geq V_i, \forall i$ and $\epsilon < \frac{1}{8\ln m}$. For every $k' \in [m/\epsilon^3, k]$, there exists $S_{k'} \subseteq S_k$ of size $k'$, such that,*

$$f_i(S_{k'}) \geq (1 - \epsilon)\Big(\frac{k' - m/\epsilon^3}{k - m/\epsilon^3}\Big)V_i, \forall i.$$

*Proof.* We restrict our ground set of elements to $S_k$ and let $S_1$ be a subset of size at most $m/\epsilon^3$ such that $f_i(e|S_1) < \epsilon^3 V_i, \forall e \in S_k \backslash S_1$ and $\forall i$ (recall, we discussed the existence of such a set in Section 2.1, Stage 1). The rest of the proof is similar to the proof of Lemma 10. Consider the point $\mathbf{x} = \frac{k' - |S_1|}{k - |S_1|}\mathbf{x}_{S_k \backslash S_1}$. Clearly, $|\mathbf{x}| = k' - |S_1|$, and from Corollary 9, we have $F_i(\mathbf{x}|\mathbf{x}_{S_1}) \geq \frac{k' - |S_1|}{k - |S_1|} F_i(\mathbf{x}_{S_k \backslash S_1}|\mathbf{x}_{S_1}) = \frac{k' - |S_1|}{k - |S_1|} f_i(S_k \backslash S_1 | S_1) \geq \frac{k' - |S_1|}{k - |S_1|}(V_i - f_i(S_1)), \forall i$. Finally, using swap rounding Lemma 1, there exists a set $S_2$ of size $k' - |S_1|$, such that $f_i(S_1 \cup S_2) \geq (1 - \epsilon)\frac{k' - |S_1|}{k - |S_1|}V_i, \forall i$.

$\square$

**Theorem 8.** *For $k' = \frac{m}{\epsilon^4}$, choosing $k'$-tuples greedily w.r.t. $h(.) = \min_i f_i(.)$ yields approximation guarantee $(1 - 1/e)(1 - 2\epsilon)$ for $k \to \infty$, while making $n^{m/\epsilon^4}$ queries.*

*Proof.* The analysis generalizes that of the standard greedy algorithm ([NW78, NWF78]). Let $S_j$ denote the set at the end of iteration $j$. $S_0 = \emptyset$ and let the final set be $S_{\lfloor k/k' \rfloor}$. Then from Theorem 7, we have that at step $j + 1$, there is some set $X \in S_k \backslash S_j$ of size $k'$ such that

$$f_i(X|S_j) \geq (1 - \epsilon)\frac{k' - m/\epsilon^3}{k - m/\epsilon^3}\big(V_i - f_i(S_j)\big), \forall i.$$

To simplify presentation let $\eta = (1 - \epsilon)\frac{k' - m/\epsilon^3}{k - m/\epsilon^3}$ and note that $\eta \leq 1$. Further, $1/\eta \to \infty$ as $k \to \infty$ for fixed $m$ and $k' = o(k)$. Now, we have for every $i$, $f_i(S_{j+1}) - (1 - \eta)f_i(S_j) \geq \eta V_i$. Call this inequality $j + 1$. Observe that inequality $\lfloor k/k' \rfloor$ states $f_i(S_{\lfloor k/k' \rfloor}) - (1 - \eta)f_i(S_{\lfloor k/k' \rfloor - 1}) \geq \eta V_i, \forall i$. Therefore, multiplying inequality $\lfloor k/k' \rfloor - j$ by $(1 - \eta)^j$ and telescoping over $j$ we get for every $i$,

$$
\begin{aligned}
f_i(S_{\lfloor k/k' \rfloor}) &\geq \sum_{j=0}^{\lfloor k/k' \rfloor - 1} (1 - \eta)^j \eta V_i \\
&\geq \big(1 - (1 - \eta)^{\lfloor k/k' \rfloor}\big)V_i \\
&\geq \big(1 - (1 - \eta)^{\frac{1}{\eta}\eta\lfloor k/k' \rfloor}\big)V_i \\
&\geq \beta(\eta\lfloor k/k' \rfloor)V_i \quad \geq (1 - 1/e)(\eta\lfloor k/k' \rfloor)V_i.
\end{aligned}
$$

Where we used (1) for the last inequality. Let $\epsilon = \sqrt[4]{\frac{m}{k'}}$, then we have,

$$\eta\lfloor k/k' \rfloor \geq (1 - \epsilon)\frac{1 - m/k'\epsilon^3}{1 - m/k\epsilon^3}\Big(1 - \frac{k'}{k}\Big) \geq \frac{\Big(1 - \sqrt[4]{\frac{m}{k'}}\Big)^2}{1 - \frac{1}{k}\sqrt[4]{\frac{m}{(k')^3}}}\Big(1 - \frac{k'}{k}\Big)$$

As $k \to \infty$ we get the asymptotic guarantee $(1 - 1/e)\Big(1 - \sqrt[4]{\frac{m}{k'}}\Big)^2 = (1 - 1/e)(1 - \epsilon)^2$.

$\square$