[Reviews · NeurIPS 2018]

Reviewer 1



The paper does submodular maximization under a cardinality constraint, when there are several functions, and we want to be close to the optimal on the minimal one. This is a natural problem, and they get the optimal 1- 1/e result. From an algorithmic perspective, they use continuous greedy and maximize the multilinear extension of several functions at once. It's not a completely new algorithm, but they do have some fine print. In the experiments section, I think your baselines were too naive. Greedy doesn't even look at the functions, and running a bi criteria algorithm without letting it pick enough elements is bound to be bad.

Reviewer 2



The paper studies a robust formulation of maximizing a submodular function subject to a cardinality constraint. Specifically, the goal is to maximize the minimum of m submodular functions f_1,..., f_m subject to picking at most k elements. When there is no restriction on k and m, the problem can be hard to approximate. The paper focuses on the regime where m << k and in this regime, it is possible to avoid the hardness and obtain approximation solution. The algorithm is inspired by previous work by CCPV'07. Previously the algorithm needs to guess the elements with large value because rounding only works when all elements have small values. Furthermore, previous work focused on getting polynomial time and uses a slow continuous greedy approach. To address the first problem, the paper focuses on a special case m<

Reviewer 3



The paper proposes three efficient multiobjective submodular maximization algorithms. I like this paper, and has no clear objection to the paper. The techniques (using one-pass procedure for picking the heavy elements, separating MWU and the continuous greedy) are simple and effective, and seem to be applied for more general problems. [minor] - l.21: The first paragraph (many well known ...) is arguable. There are many non-submodular combinatorial optimization problems. - l.60: the approximation ratio of multilinear extension-based method must be (1 - 1/e) - \epsilon. - l.89: Please write the complexity of (1 - 1/e) approx algorithm explicitly. I think it already gives significant improvement from n^{O(1/eps)} to O(poly(n, 1/eps)), and the actual bottleneck of Algorithm 1 is this factor (usually it exceeds n^{100}). - l.129: it wil be better to write x = (x_1, ..., x_n) instead of x = {x_1, ...,x_n}. [very minor] - l.129: I guess you used ... for the display math. As in https://nips.cc/Conferences/2018/PaperInformation/StyleFiles it should be avoided because it breaks line numbering.